# Bell correlations between spatially separated pairs of atoms

D.K. Shin [1], B.M. Henson[1], S.S. Hodgman[1], T. Wasak [2], J. Chwedeńczuk [3] & A.G. Truscott[1]*

Bell correlations are a foundational demonstration of how quantum entanglement contradicts the classical notion of local realism. Rigorous validation of quantum nonlocality have only been achieved between solid-state electron spins, internal states of trapped atoms, and photon polarisations, all weakly coupling to gravity. Bell tests with freely propagating massive particles, which could provide insights into the link between gravity and quantum mechanics, have proven to be much more challenging to realise. Here we use a collision between two Bose-Einstein condensates to generate spin entangled pairs of ultracold helium atoms, and measure their spin correlations along uniformly rotated bases. We show that correlations in the pairs agree with the theoretical prediction of a Bell triplet state, and observe a quantum mechanical witness of Bell correlations with $6\sigma$ significance. Extensions to this scheme could find promising applications in quantum metrology, as well as for investigating the interplay between quantum mechanics and gravity.

[1] Research School of Physics, Australian National University, Building 60, Canberra, ACT 2601, Australia. [2] Max Planck Institute for the Physics of Complex Systems, Nöthnitzer Straße 38, 01187 Dresden, Germany. [3] Faculty of Physics, University of Warsaw, ul. Pasteura 5, PL-02-093 Warszawa, Poland. *email: andrew.truscott@anu.edu.au

The basic scheme of a Bell test, originally proposed by John Bell[1], involves measuring correlated detector events between a pair of spin-$\frac{1}{2}$ particles arriving at spatially separated detectors $A$ and $B$. Bell envisioned using an entangled singlet state $|\Psi^-\rangle = (|\uparrow\rangle_A \otimes |\downarrow\rangle_B - |\downarrow\rangle_A \otimes |\uparrow\rangle_B)/\sqrt{2}$, where the states $|\uparrow\rangle$ and $|\downarrow\rangle$ are the orthogonal spin states for each particle. The particles' spins are rotated and measured independently to determine the pairwise correlations for various measurement configurations. If the observed correlation violates the so-called Bell inequality, then any description of the system will be incompatible with local realism[1,2], which requires the particles to be in a defined state at all times and not to communicate faster than the speed of light. A violation of the Bell inequality, termed Bell nonlocality, is however allowed by quantum mechanics as a consequence of entanglement between the pairs. Aside from its importance in fundamental physics, entanglement forms the basis for a wide range of quantum information processes[3], such as quantum teleportation, as well as quantum metrology[4,5].

In photonic Bell tests, the polarisation-entangled pairs are commonly generated via spontaneous parametric down-conversion (SPDC), and polarisation (analogous to spin) rotators are implemented by waveplates[6,7]. While the earliest experiments with photons[8,9] contained several loopholes that allowed for interpretations consistent with local realism[10], more recent experiments have succeeded in closing the most significant loopholes by simultaneously achieving space-like separation and almost ideal detection efficiency[6,7,11,12]. The first Bell tests using massive particles were performed by measuring spin correlations between high energy particles from nuclear decay[13,14], while the recent loophole-free experiments investigated electron spins in nitrogen-vacancy centres[11] and internal states of trapped atoms[12], where entanglement in both systems was prepared across a large distance via entanglement swapping[15]. However, such schemes rely on entanglement between internal degrees of freedom, and are unable to be extended to demonstrate entanglement in the motional degrees of freedom. This drives a fundamental motivation for investigating motional entanglement in massive systems for a possible observation of gravitational decoherence, where fluctuations of the gravitational field cause a continuous spontaneous collapse of the wavefunction at a rate, which increases with the mass of quantum matter[16].

One promising experimental system for demonstrating such entanglement is ultracold atoms[17,18]. Indeed, many characteristic quantum effects have been realised recently in ultracold atomic systems ranging from the generation of non-classical atomic pairs[19], Hong-Ou-Mandel interference[20], and the observation of spatially separated entanglement[21] and Einstein–Podolsky–Rosen steering (a stronger form of quantum nonlocality[22]) within an ensemble[23] and over a spatial separation[24,25]. A quantum mechanical (QM) witness of many-body Bell correlations has also been observed in the collective spin of a Bose–Einstein condensate (BEC)[26]. The recent efforts in the creation of such macroscopic entanglement in cold atomic ensembles have opened up applications in quantum-enhanced metrology[27]. A promising progress on the test for momentum entanglement in an atomic pair was also recently reported[28].

In this paper, we report on the generation and detection of a Bell correlation witness between the atomic spins across a spatially separated pair of metastable helium (He*) atoms. The experiment consists of the three essential components necessary to realise a Bell test: a correlated atomic pair source, a rotation of the spins of both atoms corresponding to an independently configurable measurement basis, and the momentum and spin resolved single-particle detection necessary for evaluating pair correlations. Each stage is described in detail below and shown schematically in Fig. 1. Briefly, the pair source is the binary

scattering product from a collision between the two oppositely spin-polarised BECs (Fig. 1a), which naturally separate in time. The spins of both atoms in each pair are then rotated by the same angle (Fig. 1b) followed by a measurement of their momentum and spin (Fig. 1c). Over many experimental runs we extract the correlations between the spins, which show excellent agreement with the predictions of quantum mechanics, and witness the Bell correlations in our system. We experimentally demonstrate this

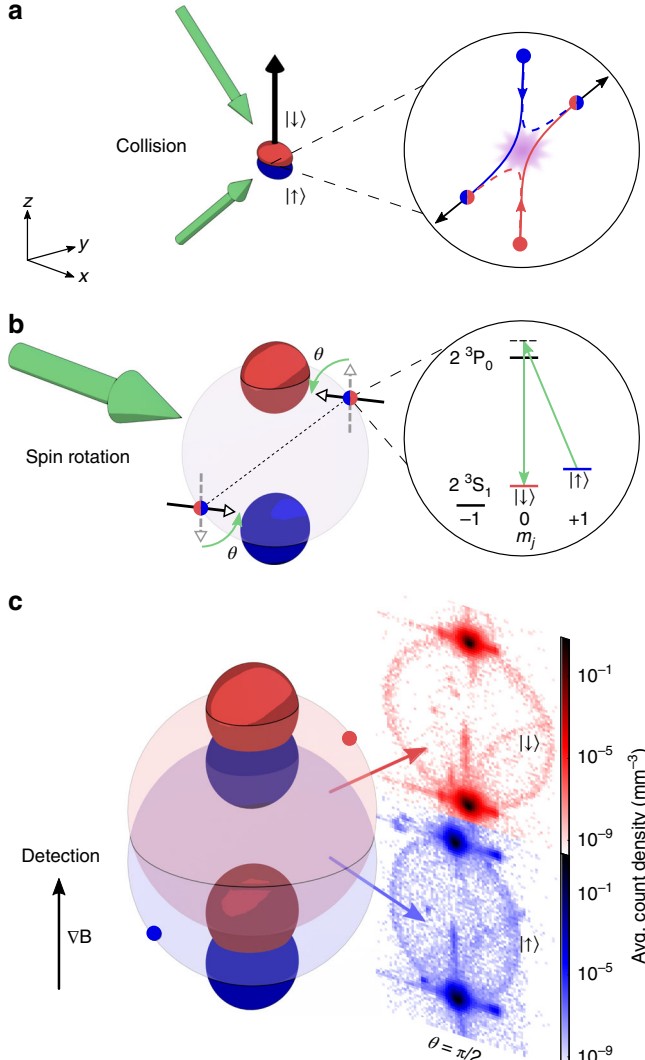

**Fig. 1** Experimental schematic for the generation and detection of entangled pairs of atoms. **a** A Raman pulse (two green arrows) initiates the collision of oppositely spin-polarised BECs (coloured ellipsoids), which scatter pairs of atoms into opposite momenta and entangled in spin as $|\Psi^+\rangle$ (see right inset). Spin-↑ (↓) state is labelled blue (red) and correspond to the $m_J = 1$ (0) state of He*. **b** The scattered pairs form a spherical shell as antipodal points in momentum, where the BECs lie on the two poles along the collision axis. Pairs spatially separate in time at which point each individual atom's spin (indicated by hollow-headed arrows) is rotated by an angle $\theta$ using co-propagating Raman beams (green arrow). The Raman transition level scheme is shown on the right (see main text for details). **c** An applied magnetic field gradient spatially separates the atoms by spin, which are detected with single-atom precision after 416 ms free-fall with full 3D momentum and spin resolution. The images on the right show the atom count density (averaged over 1000 shots) in the $zx$-plane when the spins were rotated uniformly by $\theta = \pi/2$. Spin correlations between the back-to-back scattered pairs exhibit quantum nonlocality

Bell correlation witness with a significant spatial separation of $\approx 0.1$ mm across the entangled pairs of atoms, and determine the form of quantum entanglement using a quantum state tomography technique operationally akin to the Bell test. The experimentally observed two-particle correlation, under the range of rotation angles, agrees with the predictions of a Bell triplet state, exhibits Bell correlation witness, and furthermore reveals that the atom pairs are suitable for demonstrating quantum nonlocality in a CHSH-type Bell test[29].

## Results

**Collision induced entanglement of atom pairs**. Our experiment starts with a magnetically trapped BEC of helium-4 atoms in the long-lived metastable state $2^3S_1$ (see "Methods" section for details). The atomic sublevels $|J = 1, m_J = 1\rangle = |\uparrow\rangle$ and $|J = 1, m_J = 0\rangle = |\downarrow\rangle$ form the qubit subspace (see level diagram in Fig. 1b), with the atoms initially fully spin-polarised in the $|\uparrow\rangle$ state. Following trap switch-off, a $\pi/2$-pulse from a two-photon stimulated Raman process via the $\lambda = 1083$ nm $2^3S_1 \rightarrow 2^3P_0$ transition (see Fig. 1b) simultaneously flips half of the atoms' spin to $|\downarrow\rangle$ and imparts a velocity of $\sim 120$ mm/s along the $z$-axis, opposite to gravity (see Fig. 1a). In the centre of mass frame, the two condensates split apart at $v_r \approx \pm 60$ mm/s and spontaneously scatter atoms into correlated pairs of opposite momentum and spin via binary $s$-wave collisions, forming a uniformly distributed spherical halo in momentum space with radius $k_r = 2\pi/\sqrt{2}\lambda$ (the $1/\sqrt{2}$ factor is due to 90° crossing angle of the Raman beams)[30]. In an analogy to hyperentangled photon pairs (entangled in both polarisation and momentum) generated by SPDC[31], the oppositely spin-polarised collision of BECs entangles the atom pairs in spin (see inset of Fig. 1a) as well as in momentum, from the conservation of total angular ($m_J$) and linear momentum. With the momenta of each pair given by $(\mathbf{k}, -\mathbf{k} = A, B)$, the state is symmetric under exchange of labelling by momentum and anti-correlated in spin in the original basis. Bogoliubov scattering theory predicts that the state of the pair is the archetypal Bell triplet

$$|\Psi^+\rangle = (|\uparrow\rangle_A \otimes |\downarrow\rangle_B + |\downarrow\rangle_A \otimes |\uparrow\rangle_B)/\sqrt{2} \qquad (1)$$

(See Supplementary Note 1). Such a state is maximally entangled, useful in various quantum information tasks[3], and, more importantly to this work, a viable candidate for demonstrating nonlocality[18].

**Uniform spin rotation and single-atom detection**. Following the collision pulse, the scattering halo evolves freely in a uniform magnetic field of $\sim 0.5$ G for $t_{sep} = 0.8$ ms (see Fig. 1b). The halo expands spherically such that each entangled pair, located at diametrically opposite regions of the halo, is spatially separated by $d_{sep} \approx 0.1$ mm. Larger separation distances cause a non-uniform evolution of the triplet states across the halo due to stray magnetic fields in our experiment[32]. A pair of co-propagating Raman beams (see Fig. 1b) that are wider than the size of the halo by over an order of magnitude provide a uniform rotation corresponding to

$$\hat{R}_y(\theta) = \exp\left(-i\frac{\theta}{2}\hat{\sigma}_y^{(A)}\right) \otimes \exp\left(-i\frac{\theta}{2}\hat{\sigma}_y^{(B)}\right), \qquad (2)$$

where $\hat{\sigma}_y^{(A)}$ and $\hat{\sigma}_y^{(B)}$ represent the $y$-component of Pauli matrices for spins at $A$ and $B$, while imparting no net momentum change to the atoms. The rotation is independent of the atom's momentum and position, and is applied to the whole atomic ensemble, with the rotation angle $\theta$ controlled by the Raman pulse duration. A key feature of the $|\Psi^+\rangle$ state is that it is not

rotationally invariant under a uniform rotation of both atoms in the pair by a single angle $\theta$, which enables us to measure the entanglement of the state.

Immediately after the rotation pulse, a magnetic field gradient is applied in the $z$-direction (see Fig. 1c). This projects the atoms into the $\hat{\sigma}_z$ eigenstates $\{|\uparrow\rangle, |\downarrow\rangle\}$ via the Stern–Gerlach (SG) effect, separating the two spin states in the vertical direction. Since only the $m_J = 1$ state has a non-zero magnetic moment, only $|\uparrow\rangle$ feels a magnetic force, causing the state to spatially separate from $|\downarrow\rangle$ atoms at the detector and allowing state-resolved detection.

The atoms then fall under gravity onto a microchannel plate–delay-line detector (MCP–DLD) located 0.848 m below the trap centre, which allows the crucial part of this experiment: a single-atom detection with full 3D resolution[33]. The 3D momentum $\mathbf{k}$ of each atom is reconstructed from the spatial positions (2D) and arrival time as recorded by the MCP–DLD, while the $m_J$ state is distinguished from the large separation between the arrival time of the different spin states due to the SG effect (see Supplementary Note 2). Fig. 1c shows a typical image from an average of 1000 experimental shots, displaying two completely separated halos, when a $\pi/2$-rotation was applied to evenly mix spins. Atoms in the $|\uparrow\rangle$ state form the lower halo, which is slightly non-spherical due to inhomogeneity in the magnetic field gradient causing a spatially dependent force around the halo (see Fig. 1c). Such distortion corresponds to a misalignment of the ideal back-to-back pairing in momentum and is removed in the data analysis (see "Methods" section). Since the $m_J = 0$ states are unaffected by magnetic fields, the $|\downarrow\rangle$-halo maintains the $s$-wave spherical shell shape at the detector (see the upper halo in Fig. 1c).

**Correlations in the atomic pair source**. To characterise the two-state scattering halo we look at two-particle correlations between atoms on opposite sides of the halo with either parallel or anti-parallel spin-pairing, given by

$$g_{ij}^{(2)}(\Delta\mathbf{k}) = \frac{\sum_{\mathbf{k}\in V}\langle\hat{n}_{\mathbf{k},i}\hat{n}_{-\mathbf{k}+\Delta\mathbf{k},j}\rangle}{\sum_{\mathbf{k}\in V}\langle\hat{n}_{\mathbf{k},i}\rangle\langle\hat{n}_{-\mathbf{k}+\Delta\mathbf{k},j}\rangle}, \qquad (3)$$

where $i, j \in \{\uparrow, \downarrow\}$ denote spin states, $\hat{n}_{\mathbf{q},m}$ the number of atoms with momentum $\mathbf{q}$ and spin $m$, and $V$ the volume in momentum space occupied by the $s$-wave scattering halo[34] (shown schematically in Fig. 2a). Fig. 2b shows an experimentally measured correlation function for scattering halo with no rotation, where the large peak at $\Delta\mathbf{k} = 0$ (amplitude) is indicative of strong correlations between atoms in different spin states on opposite sides of the halo. The amplitude of $g_{ij}^{(2)}$ in a spontaneous scattering halo is set by the mode occupancy $n$, given by the average number of atoms in a scattered mode, with each mode having a volume approximately that of the source condensate in momentum space[34]. The correlation amplitude is inversely proportional to the mode occupancy for a spontaneous pair source[34], which we experimentally tune by varying the starting number of atoms in the BEC prior to the collision. This relationship is verified in Fig. 2c, confirming that our pair source behaves as expected. Furthermore, the inverse proportionality is consistent with the predictions of Bogoliubov theory, which describes the pair-scattering process in the low-gain regime[34]. Importantly, we are able to reach correlation amplitudes of $\sim 60$, although due to signal to noise considerations we actually operate in a regime of $g^{(2)} = g_{\uparrow\downarrow}^{(2)}(\Delta\mathbf{k} = 0) \approx 30$, an amplitude sufficient to demonstrate a violation of a Bell inequality in the QM description[18]. The corresponding average mode occupancy in the scattering halo of $\sim 0.03$ means we are operating in the low-gain regime, where the

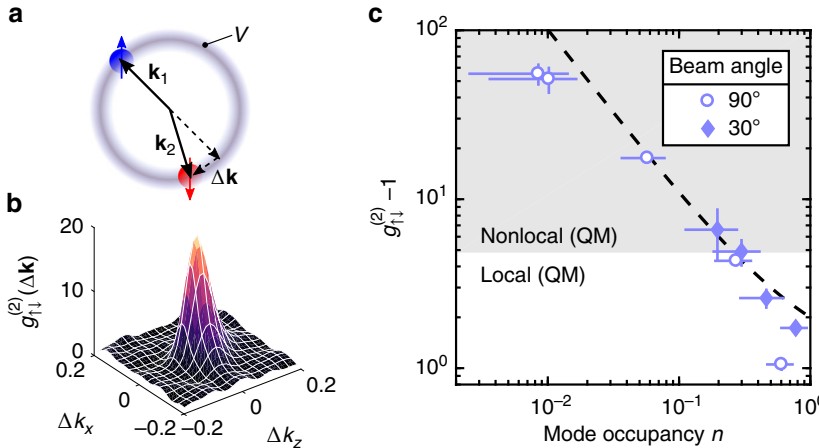

**Fig. 2** Tunable momentum-spin anti-correlated atomic pair source. **a** Schematic of atoms with momentum and spin degrees of freedom in the scattering sphere. A 2D planar slice in momentum space is taken for simplicity. **b** Two-body cross-correlation function in momentum-spin $g_{\uparrow\downarrow}^{(2)}$ from the un-rotated pair source with an average mode occupancy of $n = 0.058(2)$, averaged over 2100 experimental runs. **c** The dependence of the degree of anti-correlation on the average halo mode occupancy. Two different angles between the Raman beams (30° and 90°) were used to produce the experimental data. The dashed line depicts the theoretical prediction. Demonstration of Bell nonlocality strictly requires $g_{\uparrow\downarrow}^{(2)} > 3 + 2\sqrt{2}$ (shaded region), based on a fully QM prediction for the violation of CHSH inequality[18]. All error bars indicate the standard error in the mean

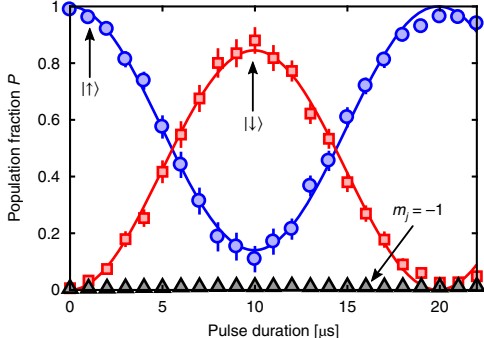

**Fig. 3** Coherent control of atomic spin in a metastable helium scattering halo. Rabi oscillation in the population fraction from a single rotation pulse with an amplitude of 0.85(4) and effective Rabi frequency $\Omega' = 2\pi \cdot 50.3(3)$ kHz. The Larmor precession frequency of the atomic spins is $\Omega_L \approx 2\pi \cdot 1.4$ MHz for a field of $\sim 0.5$ G. The error bars of the data points indicate statistical variations around different regions in the scattering sphere and the solid lines are sine fits to data

dominant contribution to the halo comes from scattering of single pairs. In a single run of the experiment which takes $\approx 30$ s, this resulted in an average of $\approx 8.3$ atoms detected in the scattering halos, and thus a joint detection rate $\approx 2$ pairs/min at $\approx 10\%$ detection efficiency.

To characterise the spin rotation, a $|\uparrow\rangle$-polarised scattering halo was initially prepared from a Raman sequence similar to our previous work to generate $m_J = 0$ halos (see refs. [34,35]). The atoms are then rotated by $\hat{R}_y(\theta)$ and the Rabi oscillations are observed (see Fig. 3) with negligible coupling to the $m_J = -1$ state.

By rotating the two-state halo and measuring the resulting correlations between atoms in each state, a two-body correlator

$$\mathcal{B}(\theta) = \left\langle \hat{\sigma}_z^{(A)} \hat{\sigma}_z^{(B)} \right\rangle_\theta = \frac{g_{\uparrow\uparrow}^{(2)} + g_{\downarrow\downarrow}^{(2)} - g_{\uparrow\downarrow}^{(2)} - g_{\downarrow\uparrow}^{(2)}}{g_{\uparrow\uparrow}^{(2)} + g_{\downarrow\downarrow}^{(2)} + g_{\uparrow\downarrow}^{(2)} + g_{\downarrow\uparrow}^{(2)}} \quad (4)$$

is obtained, where the $\theta$ subscript denotes the average in the rotated state (see Supplementary Note 1 for a derivation, and Supplementary Note 3 for the individual measurement of $g_{ij}^{(2)}$

terms). The correlator thus obtained (integrated over all scattering $\mathbf{k}$-modes) treats multiple scattered pairs in a single halo as parallel realisations of the same state in a single shot of the experiment. This embodies the inherent advantage in the rate of data acquisition by using the highly multimode scattering halo as a pair source, as opposed to few-mode counterparts, such as a twin-beam[28]. We have verified that the multiply scattered pairs are indeed identical in the spin degree-of-freedom, by showing that there is no scattering angle-dependence of the two-particle correlation functions localised in momentum[32].

The experimentally determined two-body correlator (Eq. (4)) is displayed in Fig. 4a, with the result showing an excellent agreement with the theoretical prediction $\mathcal{B}(\theta) = -\cos 2\theta$ for the Bell triplet state $|\Psi^+\rangle$. This is the first strong indication that the two atoms are strongly entangled.

**Quantum correlations**. To prove the non-classical properties of our two-atom system, we first show that the pairs are entangled. Note that for all non-entangled states, the maximum range of the correlator (Eq. (4)) is bounded by unity (see Supplementary Note 1)

$$\mathcal{S}(\theta, \theta') = |\mathcal{B}(\theta) - \mathcal{B}(\theta')| \leq 1. \quad (5)$$

We detect a clear violation of this bound in Fig. 4b, which proves the system is entangled—a necessary ingredient for the violation of a Bell inequality for any quantum system.

Since we have shown that the atomic spins are vector quantities under rotation (see Fig. 3 and Supplementary Note 2), we can now exclude a wide class of local hidden variable theories. Violation of the inequality

$$\mathcal{S}\left(\theta, \theta + \frac{\pi}{2}\right) = \left|\mathcal{B}(\theta) - \mathcal{B}\left(\theta + \frac{\pi}{2}\right)\right| \leq \sqrt{2}, \quad (6)$$

from two complementary measurements certifies the exclusion of situations in which one subsystem gives binary outcomes, whereas the second consists of a vector quantity (see Supplementary Note 1). This is violated in our system, as shown in Fig. 4b, since for $\theta = 0$ we observe $\mathcal{S}(0, \pi/2) = 1.77(6)$.

Finally, we compare our results with the predictions of Bogoliubov theory applied to the scattering process (see Supplementary Note 1), which has been tested for a wide range

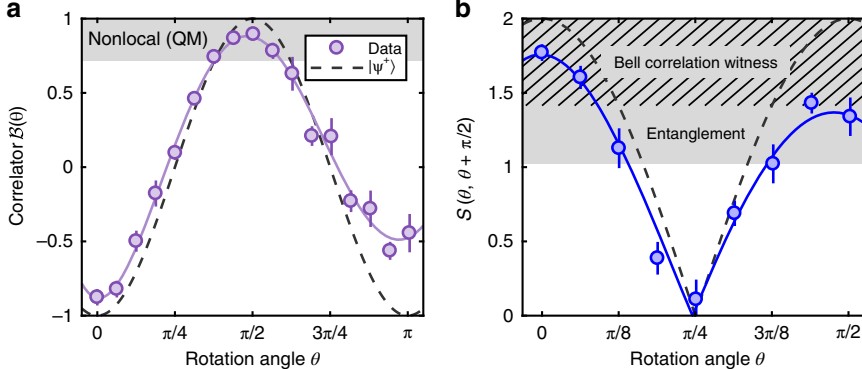

**Fig. 4** Non-classical correlations in scattering halos. **a** A strong correlation $\mathcal{B}$, exceeding $1/\sqrt{2}$ (shaded region), signals the potential to observe the violation of a Bell inequality based on a quantum mechanical model. The QM prediction of the violation was observed by $\mathcal{B}(\pi/2) = 0.90(1)$ with a significance of $18\sigma$ from 18,000 experimental runs when the spins were rotated by $\pi/2$. The dashed line is the theoretical prediction for the Bell triplet $|\Psi^+\rangle$ and the solid line is a guide to the eye, for both sub-figures. **b** Direct observation of non-classicality in the pair correlations based on the QM witness of Bell correlations $\mathcal{S}$. Correlations lying in the shaded and hatched regions indicate the presence of quantum entanglement and Bell nonlocality, respectively. $6\sigma$ violation of the Bell witness inequality (Eq. (6)) is observed by $\mathcal{S}(0, \pi/2) = 1.77(6)$. All error bars correspond to the standard error evaluated from bootstrapping

of pair production processes, ranging from the SPDC of photons to the emission of entangled atoms from colliding BECs (as in our case). In particular, this theory predicts that $\mathcal{B}(\theta) = -\cos 2\theta$. Furthermore the Bogoliubov model provides a necessary condition for the violation of the CHSH inequality in our system[29], namely $|\mathcal{B}(\theta)| > \frac{1}{\sqrt{2}}$. These are both confirmed by the experimental data in Fig. 4a.

## Discussion

This work shows that a collision of BECs generates pairwise entanglement. An extension of the scheme to a collision involving atoms of different mass (rather than spin), for instance $^3$He and $^4$He atoms, would allow the generation of isotopic entanglement in the scattered pairs. This could form the basis for testing the weak equivalence principle with quantum proof masses[36]. In addition, the spin entangled pairs generated in this work are useful for measuring the magnetic field gradient along all scattering directions, based on the coherent mixing between $|\Psi^\pm\rangle$ states when the energy splittings (i.e. Zeeman shift from local magnetic field) are different for each particle. This particular form of entanglement causes the mixing dynamics to be completely insensitive to the common-mode level of the background field[37], and can in principle achieve a measurement precision at the quantum limit[5].

In conclusion, we have demonstrated the creation and coherent control of entangled pairs of He* atoms, obtained from an $s$-wave collision halo generated from BECs in two different spin states. We have characterised the correlations between the pairs and have shown that they are sufficiently stronger than classical correlations to exhibit a QM witness of Bell correlations, and to demonstrate Bell nonlocality with further extensions. This was demonstrated by rotating the spin of each atom by a variable angle using a Raman transition. The spin and momentum resolved single-atom detection is implemented by applying a magnetic field gradient during time-of-flight. Future extensions could implement independent rotations to each entangled pair by illuminating each hemisphere of the scattering halo with separately controlled pulses of top-hat shaped Raman beams. Such near-future extension will enable a Bell test almost identical to the original proposal[1], and will exclude even those local-realist models which are incompatible with quantum mechanics. Moreover, this system could be further extended to a test of hyper-nonlocality with massive

particles[38] by simultaneously demonstrating nonlocal momentum correlations in the scattering halo[17]. Furthermore, the strong degree of quantum correlation as exhibited by the nonlocality, and counter-propagating nature of the atomic pairs studied here could find promising applications in various tasks of quantum metrology[18,27], and be extended for a quantum test of general relativity[36].

## Methods

**Experimental apparatus and procedure**. The He* BEC is initially prepared in the $m_J = 1$ state in a bi-planar quadrupole Ioffe configuration magnetic trap as explained in our previous papers[35,39], with harmonic frequencies of $(\omega_x, \omega_y, \omega_z)/2\pi \approx (15, 25, 25)$ Hz. The magnetic trap is switched-off abruptly, here denoted as time $t = 0$, from which it takes $\sim 2$ ms for the magnetic field to stabilise to a uniform field of $\mathbf{B}_0 \approx 0.5[(\mathbf{e}_x + \mathbf{e}_z)/\sqrt{2}]$ G, which splits the degeneracy in spin by $f_\updownarrow = g\mu_0 B \approx 1.4$ MHz and is maintained throughout until the SG sequence.

At $t = 3$ ms the $\pi/2$ collision Raman pulse for creating $|\Psi^+\rangle$-pairs, lasting $\sim 10$ μs, is applied from two 90°-crossed laser beams $L_1/L_2$, propagating along the $\mathbf{e}_{1/2} = (\mathbf{e}_x \pm \mathbf{e}_z)/\sqrt{2}$ directions, and $\sigma^-/\pi$-polarised with respect to the quantisation axis defined by $\mathbf{B}_0$, respectively. Each beam's optical frequencies were far-detuned from the $2^3S_1 \rightarrow 2^3P_0$ transition by $\Delta \approx 3$ GHz such that $\Delta/\Gamma \approx 2000 \gg 1$, making the single-photon absorption rate negligible. A single-photon recoil is $\hbar k_0 = 2\pi\hbar/\lambda$, where $\lambda = 1083.20$ nm.

The atoms evolve freely in the stabilised magnetic field $\mathbf{B}_0$ for 0.8 ms, at which point the scattering halo, uniformly expanding at a rate $\dot{d}_{sep} \approx 120$ mm/s (given by the recoil momenta from the two photons absorbed by the He* atoms), reaches a diameter of $d_{sep} \approx 96$ μm. The spin rotation pulse is then applied at $t = 3.8$ ms, from a second stimulated Raman transition coupled to the same transition and detuned as above, using a single beam $L_3$ that propagates along the $x$-axis. An RF-pulsed acousto-optic modulator produces a two-tones optical pulse in $L_3$, with the two frequencies fulfilling the co-propagating resonance condition for the two-photon Raman process. Furthermore, the beam was elliptically polarised with propagation along $x$-axis such that $\sigma^+$ polarisation was extinguished along $\mathbf{B}_0$, which would otherwise couple the qubit subspace to the $m_J = -1$ substate. The beam waist at the trap was $\sigma_3 \approx 1.1$ mm, an order of magnitude larger than spatial extent of the atomic ensemble at the time of rotation sequence, which provided a uniform rotation operation for all scattered atoms in the halo.

After the rotation pulse, the SG sequence is implemented by pulsing current through a large coil concentric to the $z$-axis, which selectively pushes the $m_J = +1$ atoms along the $-z$ direction. Subsequently, the atoms are detected by an 80 mm diameter microchannel plate and delay line detector located 848 mm below the trap. The free-fall duration gives the time at detection $t \approx 416$ ms, while the detector has a spatio-temporal resolution of $\sim 120$ μm $\times$ 120 μm $\times$ 3 μm[40] and a quantum efficiency of $\sim 10\%$.

**Transformation of the scattering halo**. This section provides details on the data analysis used in preprocessing the raw data from detector coordinates (position and time-of-flight) to the velocity/momentum coordinates relevant for the physical

system. Since the atoms are in free-fall for the majority of the time from the magnetic trap switch-off (see previous section on experimental procedure), the positions of atoms at the detector essentially correspond to velocities (interchangeable with momentum)[34]. As seen from Supplementary Fig. 1a, the spatial distributions of scattered atoms at the detector is aspherical for $m_J = +1$ (bottom) and spherical for $m_J = 0$ (middle). The deformation in the spatial distribution of the $m_J = +1$ scattering halo from the ideal spherical shell arises due to inhomogeneous forces from the stray magnetic field present in the vacuum chamber during free-fall. Since the accurate determination of the atomic momenta is crucial to identifying the scattered atomic pairs, a distortion correcting shape transform is applied to the raw spatial distribution of atoms ($\mathbf{r}$) to retrieve the momentum distribution ($\mathbf{k}$).

First a spatial distribution (detector coordinate) of atoms in the BECs and the scattering halo are distinguished by the internal state and the new coordinate origin defined at approximately the centre of the corresponding halo. Background atoms from the BECs, thermal fraction and miscellaneous sources other than the scattered pairs are then removed by only keeping counts lying inside the truncated spherical shell defined by $0.6 < r/r_{tof} < 1.2$ and $|r_z/r_{tof}| < 0.8$, where $r_{tof} \approx 25$ mm is the radius of the scattering halo at the detector. The resulting $\mathbf{r}$-distribution is then fitted with an ellipsoid which defines the desired smooth shape transform consisting of three orthogonal linear scalings about the centre to reduce each principal axes to unity, which can be suitably identified as the normalised momentum coordinate in the centre of mass reference frame $\mathbf{k}$ (see Fig. 2a). A final filter to remove the remaining background restricts the $\mathbf{k}$-distribution to $0.9 < k < 1.1$ and $|k_z| < 0.75$, which corresponds to the truncated momentum space $V$ investigated in this work. An additional relocation of the coordinate origin by $\vec{\mathcal{K}}$ was however necessary to centre the Gaussian profiles of $g_{ij}^{(2)}(\Delta \mathbf{k})$ at $\Delta \mathbf{k} \approx 0$, which is crucial in the implementation of Bell test as it effectively defines the two detection ports in each arm: $(\mathbf{k}, \uparrow), (\mathbf{k}, \downarrow), (-\mathbf{k}, \uparrow), (-\mathbf{k}, \downarrow)$ corresponding to the conventional $A+$, $A-$, $B+$, $B-$ detection events, respectively.

## Data availability
The data that support the findings of this study are available from the corresponding author upon reasonable request.

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

## Acknowledgements
The authors would like to thank Alain Aspect, Michael Barson, Danny Cocks, Matteo Fadel, Karen Kheruntsyan, Wolfgang Mittig, Margaret Reid, Jacob Ross and Tilman Zibold for insightful discussions. This work was supported through Australian Research Council (ARC) Discovery Project grants DP120101390, DP140101763 and DP160102337. DKS is supported by an Australian Government Research Training Program Scholarship. SSH is supported by ARC Discovery Early Career Researcher Award DE150100315. JC is supported by Project no. 2017/25/Z/ST2/03039, funded by the National Science Centre, Poland, under the QuantERA programme.

## Author contributions
D.K.S., B.M.H., S.S.H. and A.G.T. conceptualised the experiment. D.K.S. and B.M.H. acquired the data. D.K.S. analysed the data. T.W. and J.C. developed the theoretical analysis. S.S.H. and A.G.T. supervised the project. D.K.S., S.S.H., T.W., and J.C. wrote the manuscript in consultation with all authors.

## Competing interests
The authors declare no competing interests.
