## [Peer Review File · Nature Communications]

Reviewers' comments:

Reviewer #1 (Remarks to the Author):

In their manuscript entitled "Strong Bell correlations between spatially separated pairs of atoms", Shin and coworkers report an experiment aimed at producing pairs of entangled atoms.

The experiment consists in colliding two atomic Bose-Einstein condensates with different spin states, performing a global spin rotation, and measuring two-body spin correlations in the scattering halo that results from the collision. The variation of the correlations with the applied rotation angle can then be analysed to quantify the degree of entanglement.

The major claim of the paper is that entanglement is indeed observed in the atom pairs which have undergone a collision are entangled, and that the degree of entanglement would be sufficient to perform a test of a Bell inequality. Such test remains however beyond the scope of the reported experiment as it would require rotating the spin of each atom independently.

The manuscript is very well written and provides sufficient details for a member of the quantum-gas community (like me) to clearly picture the experiment. At the same time, all important ideas and concepts are introduced and explained in terms that should be easily understandable by anyone familiar with quantum mechanics.

The experimental data and its analysis are presented in a clear manner as well.

The calibration data is presented with precision in the supplemental material, which also features a step-by-step derivation of the entanglement and non-locality criteria, and the formal link with a Bell inequality.

Putting aside some difficulty with following the derivation of the non-locality criterion, I found the supplemental material of excellent quality.

All in all, the claim of a high level of entanglement seems entirely justified.

Coming to the originality of the work, it is fair to say that there are only few experiments of that sort. Entanglement between atoms has been demonstrated a number of times already, but here pairs of atoms can truly be isolated, which makes the experiment close to a text-book experiment.

To my knowledge, only one other group in the world has reported a similar type of experiment (cited as ref 24), though without proving the presence of entanglement.

I thus think that the results reported here are truly original and novel.

Still, I would have two criticisms. First, the introduction of the paper bears promises for the demonstration of entanglement in the momentum degree of freedom, which for massive particles would bridge quantum mechanics and the theory of gravity.

If it is true that the atom pairs produced in the experiment are in principle entangled jointly in spin and momentum, the authors do only measure spin correlations.

Also, they do not explain how gravity would couple to the entangled state. I do not expect a calculation of gravitational decoherence or anything like that, but a simple argument as to why gravity could potentially matter in that case.

My question is also motivated by the fact the colleagues cited in ref. 24 consider a state fully entangled in momentum (I mean $|k_1, k_2\rangle + |k_3, k_4\rangle$, instead of $|k, \uparrow\rangle + |-k, \downarrow\rangle$). Note that this other reference does not justify the potential influence of gravity either.

My second criticism would be that the paper sometimes suggests, voluntarily or not, that the obtained results would almost be equivalent to a Bell inequality test. Or that little would remain to do in order to perform a test of a Bell inequality.

This is an important point of course, because the violation of a Bell inequality implies a violation of local realism, which is even more disturbing than "just entanglement".

One example of the kind of sentence that I don't like is in the conclusion: "This yields a strength of the correlations exceeding the bound needed to violate the Bell inequality by 18σ ". The labels "Nonlocal"

in the grey-shaded area in Figs. 2 and 4 also convey the same idea.

The reason why I feel uncomfortable with this is that I think that it will be hard to extend the present experiment to perform a Bell inequality test. As already said, such test would require rotating the spin of the entangled atoms independently whereas here the rotation is made globally.

In addition to being difficult to implement, the individual addressing would also drastically reduce the statistics as it would restrict the means of averaging over a large number of entangled pairs in a single realization, as is done now. Therefore, I do not believe that violating a Bell inequality with 18σ is a realistic perspective.

Finally, regarding the publication in Nature Communications, I feel that the paper does not quite meet the criteria defined by the editors (at least as I understand them). The work is for sure of high quality and presented in a way that makes it accessible to a relatively wide audience of physicists, but I don't think that it will have a strong impact on a sufficiently wide community to warrant publication in a journal that aims at gathering truly special research results.

Reviewer #2 (Remarks to the Author):

The manuscript by Shin et al. reports on the emergence of Bell correlations between spatially separated pairs of helium atoms with opposite spin. Their experiments are in the spirit of the original Bell experiments and their results demonstrate that scattering processes in helium BECs can generate entanglement.

The paper is well written and very interesting for the field and also to a wider readership. I recommend publication in Nature Communications, after the authors have responded to my few questions and remarks:

- The scattering between the atoms creates correlated pairs of opposite momentum and spin (see e.g. Fig. 1a, right). What is the exact mechanism behind this, and why opposite spin? Does the scattering length between spin up/down atoms dominate over up-up and down-down, respectively? Add a sentence to clarify and refer to the subfigure Fig. 1a.
- Where does the $\sqrt{2}$ factor in k_r originate from? Does it result from the angle of the Raman beams? Further, does the beam waist limit the extension 0.1mm of the measured pairs or is it the free fall detection scheme or something else?
- Fig. 1: add a brief description to the left and right panels of b) (Bloch sphere and level scheme) in the caption. Add a comment to c) that these halos are recorded using free fall detection. The blue dashed circle and arrow are barely visible.
- In their discussion of the $g_{ij}(2)$ function on page 3 the authors explain that they find correlated pairs from halos as depicted in Fig. 1c. These rings are nearly solid lines, so they contain quite a lot of atoms. How many atoms are actually contained in these samples? With such data at hand, it seems straightforward to find correlations between spin up at $+k$ and spin down at $-k$, because there is a signal basically everywhere along the ring. Maybe I am missing something (related to Fig. 2c) and the correlations were obtained from data which much less events, where the determination of correlations is more obvious.

Reviewer #3 (Remarks to the Author):

The manuscript "Strong Bell correlations between spatially separated pairs of atoms" by Shin et al demonstrates the existence of Bell correlations between spatially-separated pairs of ultracold helium atoms resulting from the collision of two Bose-Einstein condensates.

This is a beautiful and nontrivial experiment. It belongs to a series of investigations exploring quantum correlations in the collision between two BECs (see works by Aspect/Westbrook and others). To the best of my knowledge, Bell correlations have never been observed before in this system. I think it can

be published in Nature Communication after taking into account my comments below.

Comments:

- The authors calculate momentum correlations by integrating over the full sphere: is it necessary? what does it happen if they integrate in only a portion of the sphere? Does a partial integration have implication on non-locality claims? The authors should comment on these aspects in the main text.

Also, I guess that the BECs at the poles are not included in the analysis. Please comment.

The detection is the crucial and winning point of this experiment: please devote one or two additional sentences to include further details about the detection.

- Fig. 2c is plotted as a function of "mode occupancy". However it is not discussed in the manuscript what is mode occupancy and how is it changed over more than two order of magnitudes. Is it post-selected from the image or tuned experimentally? Please explain.

- Implications of this work are very briefly mentioned and not discussed enough: this gives the impression that the authors are overselling their work. For instance "Extensions to this scheme would allow the study of gravitational decoherence and quantum tests of general relativity, as well as applications in quantum-enhanced metrology." What extension do the authors have in mind? This is not discussed anywhere in the manuscript. Similarly "Extensions to this scheme will allow the demonstration of EPR steering, quantum metrology, and coherent control of the He* qutrit in $^{23}\text{S}1$, as well as potential applications for the entangled pairs in quantum technologies including atom interferometry and quantum information." Please, be more detailed and avoid these obscure sentences. Either provide precise sentences and comments, or remove these overselling claims: the findings of this manuscript are interesting enough and unjustified claims do just spoil the manuscript.

- What effects does limit the visibility of Ramsey fringes in Fig. 3? Please comment.

- "The strong degree of entanglement and counter-propagating nature of the atomic pairs studied here could be used..." what does it mean "strong degree of entanglement"? Similarly, what is a "strong Bell correlation witness"? Why strong? The wording "Strong Bell correlations" is also included in the text: again, without any precise discussion, this just sounds as an attempt to oversell the work. The experiments and results are beautiful, overclaiming is not needed.

- "the quantum state of the pairs is demonstrated to be a Bell triplet state" I am not sure the authors can claim this. I would rather say that the findings agree with the prediction of a Bell triplet state. Decoherence and detection noise are necessarily present, so I do not think one can claim the creation of a precise state.

- The sentence "Aside from its importance in fundamental physics, quantum entanglement forms the basis for a wide range of applications, including quantum key distribution and quantum computing [3] to precision metrology [4]." This sentence is not very precise: 1) it is not obvious that quantum computing needs entanglement; 2) the relation between metrology and quantum entanglement was not discussed (and not reviewed) in Ref. [4].

- When mentioning "Einstein-Podolsky-Rosen (EPR) steering" with ultracold atomic systems, the authors should cite Peise et al Nat. Comm. 6:8984 (2015) - although there was no spatial separation there. In the same paragraph, when mentioning "many characteristic quantum effects" with ultra cold atoms, I suggest to cite Ref. [31] that reviews the creation of entanglement for quantum metrology.

- Li et al PRL 120, 050404 (2018) can be a useful reference

Response to the Referees

Referee 1

Comment: *In their manuscript entitled “Strong Bell correlations between spatially separated pairs of atoms”, Shin and coworkers report an experiment aimed at producing pairs of entangled atoms. The experiment consists in colliding two atomic Bose-Einstein condensates with different spin states, performing a global spin rotation, and measuring two-body spin correlations in the scattering halo that results from the collision. The variation of the correlations with the applied rotation angle can then be analysed to quantify the degree of entanglement.*

The major claim of the paper is that entanglement is indeed observed in the atom pairs which have undergone a collision are entangled, and that the degree of entanglement would be sufficient to perform a test of a Bell inequality. Such test remains however beyond the scope of the reported experiment as it would require rotating the spin of each atom independently.

The manuscript is very well written and provides sufficient details for a member of the quantum-gas community (like me) to clearly picture the experiment. At the same time, all important ideas and concepts are introduced and explained in terms that should be easily understandable by anyone familiar with quantum mechanics. The experimental data and its analysis are presented in a clear manner as well. The calibration data is presented with precision in the supplemental material, which also features a step-by-step derivation of the entanglement and non-locality criteria, and the formal link with a Bell inequality. Putting aside some difficulty with following the derivation of the non-locality criterion, I found the supplemental material of excellent quality. All in all, the claim of a high level of entanglement seems entirely justified.

Coming to the originality of the work, it is fair to say that there are only few experiments of that sort. Entanglement between atoms has been demonstrated a number of times already, but here pairs of atoms can truly be isolated, which makes the experiment close to a text-book experiment. To my knowledge, only one other group in the world has reported a similar type of experiment (cited as ref 24), though without proving the presence of entanglement. I thus think that the results reported here are truly original and novel.

Response: We thank the Referee for their very positive view of our results and paper.

Comment: *Still, I would have two criticisms. First, the introduction of the paper bears promises for the demonstration of entanglement in the momentum degree of freedom, which for massive particles would bridge quantum mechanics and the theory of gravity. If it is true that the atom pairs produced in the experiment are in principle entangled jointly in spin and momentum, the authors do only measure spin correlations.*

Response: The Referee raises a valid point and we have clarified this in the abstract by explicitly stating that we measure only spin correlations, and that the entanglement verified in this work is in the spin degree-of-freedom. The new 4th and 5th sentences of the abstract now read as follows:

Manuscript update: “Here we use a collision between two Bose-Einstein condensates to generate spin entangled pairs of ultracold helium atoms, and measure their spin correlations along uniformly rotated bases. We show that correlations in the generated pairs agree with the theoretical prediction of a Bell triplet state, and observe a quantum mechanical witness of Bell correlations with 6σ significance.”

Comment: *Also, they do not explain how gravity would couple to the entangled state. I do not expect a calculation of gravitational decoherence or anything like that, but a simple argument as to why gravity could potentially matter in that case. My question is also motivated by the fact the colleagues cited in ref. 24 consider a state fully entangled in momentum (I mean $|k_1, k_2\rangle + |k_3, k_4\rangle$, instead of $|k, \uparrow\rangle + |-k, \downarrow\rangle$). Note that this other reference does not justify the potential influence of gravity either.*

Response: We admit that our treatment of the quantum-gravity physics implications for massive systems entangled in the motional degree of freedom was somewhat hasty. To clarify this point, we have incorporated a brief discussion of one possible mechanism for gravitational decoherence, with a reference to a recent review article on the topic [1]. The argument is similar to the decoherence mechanism due to random noise: massive systems couple more strongly to the gravitational field, in which random fluctuations can directly cause the phase relationship in the motional superposition to smear out. Indeed, the reason for the elusiveness of quantum-gravity effects in laboratory experiments is due to the weak coupling strength. This can be addressed with macroscopic quantum matter (optomechanical systems) and, by the exquisite controllability and environmental isolation offered by the fundamental quantum systems, ultracold atoms (see e.g. the recent article [2]). Our updated final sentence of the 2nd paragraph of the paper, discussing the implications for quantum-gravity tests, is below:

Manuscript update: “This drives a fundamental motivation for investigating motional entanglement in massive systems for a possible observation of gravitational decoherence, where fluctuations of the gravitational field cause a continuous spontaneous collapse of the wavefunction at a rate which increases with the mass of quantum matter [1].”

Comment: *My second criticism would be that the paper sometimes suggests, voluntarily or not, that the obtained results would almost be equivalent to a Bell inequality test. Or that little would remain to do in order to perform a test of a Bell inequality. This is an important point of course, because the violation of a Bell inequality implies a violation of local realism, which is even more disturbing than “just entanglement”. One example of the kind of sentence that I don’t like is in the conclusion: “This yields a strength of the correlations exceeding the bound needed to violate the Bell inequality by 18σ ”. The labels “Nonlocal” in the grey-shaded area in Figs. 2 and 4 also convey the same idea. The reason why I feel uncomfortable with this is that I think that it will be hard to extend the present experiment to perform a Bell inequality test. As already said, such test would require rotating the spin of the entangled atoms independently whereas here the rotation is made globally. In addition to being difficult to implement, the individual addressing would also drastically reduce the statistics as it would restrict the means of averaging over a large number of entangled pairs in a single realisation, as is done now. Therefore, I do not believe that violating a Bell inequality with 18σ is a realistic perspective.*

Response: The Referee raises two important points: first, for a clear distinction of our nonlocality claims from that of a Bell inequality violation, and second, scepticism towards the feasibility for extending the current scheme to a Bell inequality test. Below, we have addressed these points in order. First, we would like to underline that Figures 2c and 4a refer to a model-based criterion for nonlocality of pairs of atoms which become spin-entangled as part of the s-wave scattering process. Indeed, without the consideration for the quantum state, these correlations are irrelevant to Bell inequalities. In order to clearly distinguish this result from the violation of a Bell inequality, we had originally used “(QM)” to label the regions for nonlocality claims. Correlations which enter this regime witness not only the entanglement but also a high degree of non-classicality that, assuming the validity of the quantum mechanics, signals Bell nonlocality. We have emphasised this point in Fig. 2 caption, as shown below.

Manuscript update: “Demonstration of Bell nonlocality strictly requires $g_{\uparrow\downarrow}^{(2)} > 3 + 2\sqrt{2}$ (shaded region), based on a fully QM prediction for the violation of a CHSH inequality [3].”

Response continued: Second, as pointed out by the Referee, to exclude all local-realist theories — even those which are not compatible with quantum mechanics — a complete Bell inequality test should be performed. We strongly believe such an extension is feasible and is a goal of our lab for the near future.

Such a scheme could proceed as follows: we will illuminate the two halves of the scattering halo (i.e. hemispheres) with separate Raman beams to apply spin-rotations independently for each atom in the pairs. Here, the experimental challenge resides in shaping a Raman beam profile so that it illuminates only the target hemisphere. For this we will use a rectangular ‘top-hat’ profile for each beam, creating a broad, uniform intensity region that illuminates the majority of each respective hemisphere. Diffraction means that the beam profile will not have a perfectly sharp edge, although edge-widths of below $\approx 10 \mu\text{m}$ can be readily achieved with commercial diffractive optics or phase masks. Therefore, for a $\approx 100 \mu\text{m}$ diameter halo, a conservative estimate of $\approx 80\%$ of atoms will be illuminated with a uniform intensity by each beam, making independent spin-rotation for hemispheres possible by varying each beam’s pulse duration. Therefore, the Bell inequality test is a feasible extension to our experiment.

Once the shaping and alignment of the rotation beams are optimised, the majority of the scattering halo will be spin-rotated independently. However, pairs located near the boundary of the two hemispheres will need to be removed from the analysis, due to the non-uniform potential (and thus spin rotation) at the edges of the Raman beams. The thickness of such exclusion region will be a combination of the wavefunction width of each atom, and the edge-width of the Raman beam. A conservative approximation predicts a loss of $\approx 20\%$ of the presently detected pairs (i.e. an exclusion of a $20 \mu\text{m}$ thick region from a halo of $100 \mu\text{m}$ diameter). Note that expanding the scattering halo further (before rotation) could also be used to decrease the relative size of the exclusion region. However, even with a much more conservative estimate of 50% of the halo pairs unable to be used, we will still only require ≈ 4 times as much data acquisition per rotation setting (e.g. the four CHSH-Bell test angles) to achieve the same statistical degree of violation as predicted here. For the 18σ point here we acquired $\approx 18,000$ shots, therefore a Bell inequality test with 18σ is a realistic perspective as an extension to our current method.

This discussion on the feasibility of the proposed Bell inequality test extension is now summarised in the conclusion of the main text:

Manuscript update: “Future extensions could implement independent rotations to each entangled pair by illuminating each hemisphere of the scattering halo with separately controlled pulses of top-hat shaped Raman beams. Such a near-future extension will enable a Bell test almost identical to the original proposal [4], and will exclude even those local-realist models which are incompatible with quantum mechanics.”

Comment: *Finally, regarding the publication in Nature Communications, I feel that the paper do not quite meet the criteria defined by the editors (at least as I understand them). The work is for sure of high quality and presented in a way that makes it accessible to a relatively wide audience of physicists, but I don’t think that it will have a strong impact on a sufficiently wide community to warrant publication in a journal that aims at gathering truly special reaserch results.*

Response: We trust that after our clarifications and the improvements made to the text, along with the very positive feedback from other two Referees, the manuscript deserves the attention of the broad community of the readers of Nature Communications.

Referee 2

Comment: *The manuscript by Shin et al. reports on the emergence of Bell correlations between spatially separated pairs of helium atoms with opposite spin. Their experiments are in the spirit of the original Bell experiments and their results demonstrate that scattering processes in helium BECs can generated entanglement.*

The paper is well written and very interesting for the field and also to a wider readership. I recommend publication in Nature Communications, after the authors have responded to my few questions and remarks:

Response: We thank the Referee for their kind words and their appreciation for the significance of our results.

Comment: *The scattering between the atoms creates correlated pairs of opposite momentum and spin (see e.g. Fig. 1a, right). What is the exact mechanism behind this, and why opposite spin? Does the scattering length between spin up/down atoms dominate over up-up and down-down, respectively? Add a sentence to clarify and refer to the subfigure Fig. 1a.*

Response: The first Raman pulse in Fig. 1a splits an original BEC into two daughter condensates: the original state with $m_J = 1$ (which we denote as “spin-up”) with zero-momentum, and one that is imparted with a momentum-kick by the stimulated Raman transition, as well as a change in internal state to $m_J = 0$ (which we denote as “spin-down”). In the centre of mass frame, these condensates mark the poles of the scattering halo in momentum-space. The individual entangled pairs are generated via binary s -wave atomic collisions between atoms from the two colliding daughter condensates. This process conserves the total m_J so that no other spin configurations (such as up-up and down-down) contribute to the scattering halo which we interrogate for correlations. The elastic s -wave scattering process ensures that pairs are scattered with opposite momenta, as the collision has to conserve momentum as well. As requested, we have clarified this mechanism in the 2nd paragraph of page 2:

Manuscript update: “. . . the oppositely spin-polarised collision of BECs entangles the atom pairs in spin (see inset of Fig. 1(a)) as well as in momentum, from the conservation of total angular m_J and linear momentum. ”

Response continued: In addition, the caption for Fig. 1a now summarises this (2nd sentence of Figure 1 caption):

Manuscript update: “(a) A Raman pulse initiates the collision of oppositely spin-polarised BECs, which scatter pairs of atoms into opposite momenta and entangled in spin as $|\Psi^+\rangle$.”

Comment: *Where does the $\sqrt{2}$ factor in k_r originate from? Does it result from the angle of the Raman beams? Further, does the beam waist limit the extension 0.1 mm of the measured pairs or is it the free fall detection scheme or something else?*

Response: Indeed, the Raman beams used to initiate the BEC collision are crossed at 90 degrees, such that the stimulated two-photon transition imparts a net momentum $\sqrt{2}$ times that of a single photon recoil $2\pi/\lambda$ to $|\downarrow\rangle$ BEC. The radius of the scattering halo is half of this, giving the desired expression. We have revised the manuscript (see 2nd paragraph of page 2) to make this clear.

Manuscript update: “In the centre of mass frame, the two condensates split apart at $v_r \approx \pm 60$ mm/s and spontaneously scatter atoms into correlated pairs of opposite momentum and spin via binary s -wave collisions, forming a uniformly distributed spherical halo in momentum space with radius $k_r = 2\pi/\sqrt{2}\lambda$ (the $1/\sqrt{2}$ factor is due to 90° crossing angle of the Raman beams) [5]”

Response continued: The spin-rotating Raman beam waist ($1/e$ radius 1.1 mm) is large enough to ensure a uniform rotation for atoms on opposite sides of the halo, even for larger halos than 0.1 mm diameter. Indeed to show that the rotation was uniform across the atom pairs, the Rabi oscillation characterisation shown in Fig. 3 was performed with a 0.36 mm diameter halo. The pair separation was limited to 0.1 mm in this experiment because when the pairs were allowed to expand further, we observed some of the entangled pairs to evolve from $|\Psi^+\rangle$ to the Bell singlet state $|\Psi^-\rangle$, due to the inhomogeneity in the magnetic field across the halo. We have further investigated the mixing dynamics between the Bell states, details of which can be found in [6]. To help clarify this to the reader, we have added the following sentence to the last paragraph on page 2:

Manuscript update: “Larger separation distances cause a non-uniform evolution of the triplet states across the halo due to stray magnetic fields in our experiment [6].”

Comment: *Fig. 1: add a brief description to the left and right panels of b) (Bloch sphere and level scheme) in the caption. Add a comment to c) that these halos are recorded using free fall detection. The blue dashed circle and arrow are barely visible.*

Response: We have followed the Referee’s suggestions and have added the following to the caption of Fig. 1:

Manuscript update:

“... each individual atom’s spin is rotated by an angle θ on the Bloch sphere (shown on the left) using co-propagating Raman beams. The Raman transition level scheme is shown on the right.”

“... which are detected with single-atom precision after 416 ms free-fall with full 3D momentum and spin resolution.”

Response continued: We have improved the visibility of Fig. 1, as pointed out by the referee. Brighter colours and solid line-styled arrows are used, while the dashed circles were removed.

Comment: *In their discussion of the $g_{ij}^{(2)}$ function on page 3 the authors explain that they find correlated pairs from halos as depicted in Fig. 1c. These rings are nearly solid lines, so they contain quite a lot of atoms. How many atoms are actually contained in these samples? With such data at hand, it seems straightforward to find correlations between spin up at $+k$ and spin down at $-k$, because there is a signal basically everywhere along the ring. Maybe I am missing something (related to Fig. 2c) and the correlations were obtained from data which much less events, where the determination of correlations is more obvious.*

Response: As the Referee observed, the 2D density images in Fig. 1c clearly depict scattering halos as almost uniformly populated rings. However, please note that these images show the density averaged over many experimental shots. The pairwise joint detection signal is obtained from single-shot results. We admit that this was not sufficiently clarified in the figure’s caption and the main text missed explicit detail for the average number of scattered atoms. To clarify this we have amended the main text and the figure caption to emphasise that the images in Fig. 1c are an average of 1000 experimental shots, and provide single-shot details of the pair generation, as follows.

Manuscript update:

Figure 1 caption:

“The images on the right show the atom count density (averaged over 1000 shots) in the zx -plane ...”

Main text, 3rd paragraph page 3:

“Figure 1(c) shows a typical image from an average of 1000 experimental shots, ...”

Main text, last sentence of page 3:

“In a single run of the experiment which takes ≈ 30 s, this resulted in an average of ≈ 8.3 atoms detected in the scattering halos, and thus a joint detection rate ≈ 2 pairs/min at $\approx 10\%$ detection efficiency. ”

Referee 3

Comment: *The manuscript “Strong Bell correlations between spatially separated pairs of atoms” by Shin et al demonstrates the existence of Bell correlations between spatially-separated pairs of ultracold helium atoms resulting from the collision of two Bose-Einstein condensates.*

This is a beautiful and nontrivial experiment. It belongs to a series of investigations exploring quantum correlations in the collision between two BECs (see works by Aspect/Westbrook and others). To the best of my knowledge, Bell correlations have never been observed before in this system. I think it can be published in Nature Communication after taking into account my comments below.

Response: We thank the referee for their very supportive comments.

Comment: *The authors calculate momentum correlations by integrating over the full sphere: is it necessary? what does it happen if they integrate in only a portion of the sphere? Does a partial integration have implication on non-locality claims? The authors should comment on these aspects in the main text.*

Response: This is an interesting observation. In our work, the joint probability for spin-spin configurations for atom pairs, identified by opposite momenta $(k, -k)$, are integrated over the whole scattering halo. This means that different scattered atom pairs $\{(q, -q) \neq (k, -k)\}$ are treated as identical copies of the same state for the evaluation of the spin correlation functions. As per the Referee’s comment, a partial/localised integration in the scattering halo serves as a powerful microscope that would reveal any scattering angle-dependent features of correlation.

We have investigated the uniformity across the halo in a separate experiment [6], where we verified that the pairs comprising the scattering halo are identical in the spin degree-of-freedom. The full integration is therefore only necessary to minimise the statistical uncertainty in our non-locality claims (maximising the size of the data set). Integrating only a portion of the scattering halo reduces the signal-to-noise ratio.

Following the Referee’s suggestion, we include a brief discussion of this point in the main text, adding the following text directly after equation (2):

Manuscript update: “The correlator thus obtained (integrated over all scattering \mathbf{k} -modes) treats multiple scattered pairs in a single halo as parallel realisations of the same state in a single shot of the experiment. This embodies the inherent advantage in the rate of data acquisition by using the highly multimode scattering halo as a pair source, as opposed to few-mode counterparts such as a twin-beam [7]. We have verified that the multiply scattered pairs are indeed identical in the spin degree-of-freedom, by showing that there is no scattering angle-dependence of the two-particle correlation functions localised in momentum [6].”

Comment: *Also, I guess that the BECs at the poles are not included in the analysis. Please comment.*

Response: Due to the possible detector saturation caused by the large flux of the BECs and the need to remove unscattered BEC atoms from the analysis, we remove all atoms lying near the poles of the sphere in the pre-processing stage of our data analysis. The spatial filter implemented is a truncated spherical shell, full details of which are given in the supplementary section: “Transformation of the scattering halo” in page 9.

Comment: *The detection is the crucial and winning point of this experiment: please devote one or two additional sentences to include further details about the detection.*

Response: We have expanded upon the originally brief mention of the single atom detection, to give further details emphasising its unique points for correlation experiments. The amended text for the detection procedure shown below is added to the 3rd paragraph on page 3:

Manuscript update: “The atoms then fall under gravity onto a microchannel plate – delay-line detector (MCP-DLD) located 0.848 m below the trap centre, which allows the crucial part of this experiment: a single-atom detection with full 3D resolution [8]. The 3D momentum \mathbf{k} of each atom is reconstructed from the spatial positions (2D) and arrival times as recorded by the MCP-DLD, while the m_J state is distinguished from the large separation between the arrival time of the different spin states due to the SG effect [SOMs]. ”

Comment: *Fig. 2c is plotted as a function of “mode occupancy”. However it is not discussed in the manuscript what is mode occupancy and how is it changed over more than two order of magnitudes. Is it post-selected from the image or tuned experimentally? Please explain.*

Response: The Referee makes a good point that we have somewhat skipped over these details, due to our previous detailed investigation of correlations and mode occupancy in scattering halos [9]. To clarify these points in the manuscript we have amended the paragraph where “mode occupancy” is first introduced (4th paragraph on page 3) to now read:

Manuscript update: “The amplitude of $g_{ij}^{(2)}$ in a spontaneous scattering halo is set by the mode occupancy n , given by the average number of atoms in a scattered mode, with each mode having a volume approximately that of the source condensate in momentum space [9]. The correlation amplitude is inversely proportional to the mode occupancy for a spontaneous pair source [9], which we experimentally tune by varying the starting number of atoms in the BEC prior to the collision. ”

Response continued: For consistency, we have also replaced all usages of “mode occupation” with “mode occupancy” in the manuscript.

Comment: *Implications of this work are very briefly mentioned and not discussed enough: this gives the impression that the authors are overselling their work. For instance “Extensions to this scheme would allow the study of gravitational decoherence and quantum tests of general relativity, as well as applications in quantum-enhanced metrology.” What extension do the authors have in mind? This is not discussed anywhere in the manuscript. Similarly “Extensions to this scheme will allow the demonstration of EPR steering, quantum metrology, and coherent control of the He* qutrit in 2^3S_1 , as well as potential applications for the entangled pairs in quantum technologies including atom interferometry and quantum information.” Please, be more detailed and avoid these obscure sentences. Either provide precise sentences and comments, or remove these overselling claims: the findings of this manuscript are interesting enough and unjustified claims do just spoil the manuscript.*

Response: As the Referee has pointed out, there were a number of claims regarding the implications of this work which cited a reference but missed an explicit discussion. In order to provide the reader with a clearer picture of the implications of our work, we have rewritten the relevant sentences flagged by the Referee so that they clearly state our intention. For clarity, we have also reduced the list of claims that we are making.

In the abstract the salient implications are summarised:

Manuscript update: “Extensions to this scheme could find promising applications in quantum metrology, as well as investigating the interplay between quantum mechanics and gravity.”

Response continued: These points are discussed in detail in a separate paragraph preceding the conclusions:

Manuscript update: “This work shows that a collision of BECs generates pairwise entanglement. An extension of the scheme to a collision involving atoms of different mass (rather than spin), for instance ^3He and ^4He atoms, would allow the generation of isotopic entanglement in the scattered pairs. This could form the basis for testing the weak equivalence principle with quantum proof masses [10]. In addition, the spin entangled pairs generated in this work is useful for measuring the magnetic field gradient along all scattering directions, based on the coherent mixing between $|\Psi^\pm\rangle$ states when the energy splittings (i.e. Zeeman shift from local magnetic field) are different for each particle. This particular form of entanglement causes the mixing dynamics to be completely insensitive to the common-mode level of the background field [6, 11], and can in principle achieve a measurement precision at the quantum limit [12]”

Response continued: And summarised once more in the conclusion:

Manuscript update: “... atomic pairs studied here could find promising applications in various tasks of quantum metrology [3, 13], and be extended for a quantum test of general relativity [10].”

Comment: *What effects does limit the visibility of Ramsey fringes in Fig. 3? Please comment.*

Response: We suspect that the non-ideal visibility of the Rabi oscillation is most likely due to a finite amount of detuning in the Raman transition present in the experiment. Operationally speaking, as long as the mixing/rotation operations (Raman process) are switchable in some way, different classes of quantum nonlocality can be tested. However, the optimal angles giving rise to the maximal violation may not be accessible with a single pulse. Since test particles are quantum mechanical, the rotation operation must be able to sweep over enough of an angle that a violation can be observed. In our case the slight reduction from the ideal visibility of the Rabi fringe is not significant enough to affect the central results of this work.

Comment: *“The strong degree of entanglement and counter-propagating nature of the atomic pairs studied here could be used...” what does it mean “strong degree of entanglement”? Similarly, what is a “strong Bell correlation witness”? Why strong? The wording “Strong Bell correlations” is also included in the text: again, without any precise discussion, this just sounds as an attempt to oversell the work. The experiments and results are beautiful, overclaiming is not needed.*

Response: The Referee makes a good point that our language usage could be tighter, in particular our usage of the description “strong” for the degree of entanglement and Bell correlation witness, which we address separately.

To clarify the meaning of strong in the context of the degree of entanglement, which in our case we use to distinguish the quantum correlations from “entanglement” and “Bell correlation witness”, we have changed the last sentence of the conclusion to now read as follows:

Manuscript update: “Furthermore, the strong degree of quantum correlation, as exhibited by the nonlocality, ...”

Response continued: We agree that our usage of the word “strong” to define Bell correlations may confuse the reader, since Bell correlation is a measurable outcome “incompatible with local realism”, and as such is a binary property which does not benefit from words such as “strong” (i.e. strong Bell correlation). Therefore, we have replaced “strong” throughout the manuscript with the statistical significance by which the corresponding inequalities were violated. Relevant sentences in the manuscript have been amended as below.

Manuscript update:

In all occurrences (including in the title) of “strong” referring to Bell correlations we have removed the word “strong”.

(In the abstract): “... observe a quantum mechanical witness of Bell correlations with 6σ significance.”

(2nd sentence of the conclusion): “... correlations between the pairs ... are sufficiently stronger than classical correlations to exhibit a QM witness of Bell correlations, and to demonstrate Bell nonlocality with further extensions.”

Response continued: However, the description “strong” belongs naturally to quantifiable correlations, such as the $g^{(2)}$, \mathcal{B} , and \mathcal{S} parameters for which the “strengths” (i.e. relatively high numerical values) are bound by different types of quantum non-locality, and as such we have left any instances of this usage.

Comment: *“the quantum state of the pairs is demonstrated to be a Bell triplet state” I am not sure the authors can claim this. I would rather say that the findings agree with the prediction of a Bell triplet state. Decoherence and detection noise are necessarily present, so I do not think one can claim the creation of a precise state.*

Response: We agree with the Referee, and have adopted their suggestion by amending the second to last sentence in the abstract:

Manuscript update: “We show that correlations in the generated pairs agree with the theoretical prediction of a Bell triplet state, ...”

Comment: *The sentence “Aside from its importance in fundamental physics, quantum entanglement forms the basis for a wide range of applications, including quantum key distribution and quantum computing [3] to precision metrology [4].” This sentence is not very precise:*

- 1) *it is not obvious that quantum computing needs entanglement;*
- 2) *the relation between metrology and quantum entanglement was not discussed (and not reviewed) in Ref. [4].*

Response:

ad 1) We thank the Referee for this information. Indeed, we have found that the role entanglement plays for all general quantum computation schemes is currently not well understood. To rectify the error we have removed this part of the statement.

ad 2) Since the original reference (Ref. [4]) ”Giovannetti et al, Quantum-Enhanced Measurements: Beating the Standard Quantum Limit, Science 306 (2004)” is admittedly written for the general readership, it contains brief explanations on entanglement-enhanced strategies to go beyond the standard quantum limit (i.e. quantum metrology). To be more specific, we have replaced the reference cited for this point to a work that applies an in-depth treatment of quantum vs classical metrology, which finds entanglement as the central resource: [12].

To address both points 1) and 2), the last sentence of the first paragraph of the paper has been slightly amended to now read:

Manuscript update: “Aside from its importance in fundamental physics, entanglement forms the basis for a wide range of quantum information processes [14], such as quantum teleportation, as well as quantum metrology [12]. ”

Comment: *When mentioning “Einstein-Podolsky-Rosen (EPR) steering” with ultracold atomic systems, the authors should cite [15] — although there was no spatial separation there. In the same paragraph, when mentioning “many characteristic quantum effects” with ultra cold atoms, I suggest to cite Ref. [31] that reviews the creation of entanglement for quantum metrology.*

Response: The Referee’s suggestions have been adopted and implemented by changing the relevant sentences in the 3rd paragraph of the paper to now read:

Manuscript update:

“... Einstein-Podolsky-Rosen (EPR) steering (a stronger form of quantum nonlocality [16]) within an ensemble [15] and over a spatial separation...”

“The recent efforts in the creation of such macroscopic entanglement in cold atomic ensembles has opened up applications in quantum-enhanced metrology [13].”

Comment: *[17] can be a useful reference*

Response: We would like to thank the Referee for informing us of the useful reference on hyper-entanglement (entanglement in more than one degree-of-freedom) and novel tests of non-locality which can be performed with such resource. The implications of this reference for our work, and a seminal paper cited within [18], have been added to the main text. The relevant additions are listed below:

Manuscript update:

2nd paragraph page 2:

“In an analogy to hyperentangled photon pairs (entangled in both polarisation and momentum) generated by SPDC [18], the oppositely spin-polarised collision of BECs entangles the atom pairs in spin (see inset of Fig. 1(a)) as well as in momentum, from the conservation of total angular (m_J) and linear momentum.”

Last paragraph:

“Such near-future extension will enable a Bell test almost identical to the original proposal [4], and will exclude even those local-realistic models which are incompatible with quantum mechanics. Moreover, this system could be further extended to a test of hyper-nonlocality with massive particles [17] by simultaneously demonstrating nonlocal momentum correlations in the scattering halo [19].”

-
- [1] A. Bassi, A. Groß and H. Ulbricht, *Class. Quantum Grav.* **34**, 193002 (2017)
 - [2] R. Howl, R. Penrose and I. Fuentes, *New J. Phys.* **21** 043047 (2019)
 - [3] T. Wasak and J. Chwedeńczuk, *Phys. Rev. Lett.* **120**, 140406 (2018)
 - [4] J. Bell, *Physics* **1** 195 (1964)
 - [5] A. Perrin, H. Chang, V. Krachmalnicoff, M. Schellekens, D. Boiron, A. Aspect, and C. I. Westbrook, *Phys. Rev. Lett.* **99**, 150405 (2007)
 - [6] D. K. Shin, J. A. Ross, B. M. Henson, S. S. Hodgman, A. G. Truscott, arxiv:1906.08958
 - [7] Pierre Dussarrat, Maxime Perrier, Almazbek Imanaliev, Raphael Lopes, Alain Aspect, Marc Cheneau, Denis Boiron, and Christoph I. Westbrook, *Phys. Rev. Lett.* **119**, 173202 (2017)
 - [8] Wim Vassen, Claude Cohen-Tannoudji, Michele Leduc, Denis Boiron, Christoph I. Westbrook, Andrew Truscott, Ken Baldwin, Gerhard Birkel, Pablo Cancio, and Marek Trippenbach, *Rev. Mod. Phys.* **84**, 175 (2012)
 - [9] S.S. Hodgman, R.I. Khakimov, R.J. Lewis-Swan, A.G. Truscott, and K.V. Kheruntsyan, *Phys. Rev. Lett.* **118**, 240402 (2017)
 - [10] Remi Geiger and Michael Trupke, *Phys. Rev. Lett.* **120**, 043602 (2018)
 - [11] D. Kielpinski, V. Meyer, M. A. Rowe, C. A. Sackett, W. M. Itano, C. Monroe, D. J. Wineland, *Science* **291**, 1013 (2001)
 - [12] Vittorio Giovannetti, Seth Lloyd, and Lorenzo Maccone, *Phys. Rev. Lett.* **96**, 010401 (2006)
 - [13] Luca Pezzé, Augusto Smerzi, Markus K. Oberthaler, Roman Schmied, and Philipp Treutlein, *Rev. Mod. Phys.* **90**, 035005 (2018)
 - [14] Ryszard Horodecki, Paweł Horodecki, Michał Horodecki, and Karol Horodecki, *Rev. Mod. Phys.* **81**, 865 (2009)
 - [15] J. Peise, I. Kruse, K. Lange, B. Lücke, L. Pezzé, J. Arlt, W. Ertmer, K. Hammerer, L. Santos, A. Smerzi and C. Klempt, *Nat. Comm.* **6**, 8984 (2015)
 - [16] H. M. Wiseman, S. J. Jones, and A. C. Doherty, *Phys. Rev. Lett.* **98**, 140402 (2007)
 - [17] Y. Li, M. Gessner, W. Li, and A. Smerzi, *Phys. Rev. Lett.* **120**, 050404 (2018)
 - [18] P. G. Kwiat, *J. Mod. Opt.* **44**, 2173 (1997)
 - [19] R. J. Lewis-Swan and K. V. Kheruntsyan, *Phys. Rev. A* **91**, 052114 (2015)

REVIEWERS' COMMENTS:

Reviewer #1 (Remarks to the Author):

The authors have convincingly replied to all issues I raised. The revised version of the manuscript is excellent. I do not object to publication in Nature Communications anymore.

Reviewer #2 (Remarks to the Author):

The authors have addressed all my remarks and questions adequately. Therefore, I recommend publication of their manuscript in Nature Communications.

Reviewer #3 (Remarks to the Author):

I thank the authors for the reply to all my comments and for the related changes in the manuscript. About the authors changes in the text, my only concern is about the reference on entanglement and metrology: the standard and relevant reference is Phys. Rev. Lett. 102, 100401 (2009), that should be cited in place (or together) with [4].. Apart this detail, I am glad to recommend the publication of this manuscript in Nature Communications.

Response to the Reviewers

Dear Editors,

We thank the Reviewers once again for their initial feedback and carefully considering our response. From this correspondence we are very glad to have received unanimously positive comments and recommendations from the Reviewers. We now present minor amendments to the manuscript addressing the remaining points from their most recent communication, as well as formatting our manuscript to comply with Nature Communications' standards.

With best regards,

Dongki Shin, Bryce M. Henson, Sean S. Hodgman, Tomasz Wasak, Jan Chwedeńczuk, and Andrew G. Truscott

Reviewer 1

Comment: *The authors have convincingly replied to all issues I raised. The revised version of the manuscript is excellent. I do not object to publication in Nature Communications anymore.*

Response: We thank the reviewer for their initial critique and the examined support after reconsidering our response.

Reviewer 2

Comment: *The authors have addressed all my remarks and questions adequately. Therefore, I recommend publication of their manuscript in Nature Communications.*

Response: We thank the reviewer for their insightful feedback and positive comments.

Reviewer 3

Comment: *I thank the authors for the reply to all my comments and for the related changes in the manuscript.*

Response: We thank the reviewer for the precise points and very supportive comments.

Comment: *About the authors changes in the text, my only concern is about the reference on entanglement and metrology: the standard and relevant reference is Phys. Rev. Lett. 102, 100401 (2009), that should be cited in place (or together) with [1].*

Response: We would like to thank the reviewer for suggesting to us the reference, which we found to be a very informative article. As the reviewer states, we also believe this result is appropriate to be cited along with the original reference discussing the role of quantum correlation to metrology. The relevant sentence in manuscript is given below.

Manuscript update: “Aside from its importance in fundamental physics, entanglement forms the basis for a wide range of quantum information processes [2], such as quantum teleportation, as well as quantum metrology [1, 3].”

Comment: *Apart this detail, I am glad to recommend the publication of this manuscript in Nature Communications.*

Response: We once again thank the reviewer for their appreciation of our work.

-
- [1] V. Giovannetti, S. Lloyd, and L. Maccone, “Quantum metrology,” *Phys. Rev. Lett.* **96**, 010401 (2006).
 - [2] R. Horodecki, P. Horodecki, M. Horodecki, and K. Horodecki, “Quantum entanglement,” *Rev. Mod. Phys.* **81**, 865–942 (2009).
 - [3] L. Pezzé and A. Smerzi, “Entanglement, nonlinear dynamics, and the Heisenberg limit,” *Phys. Rev. Lett.* **102**, 100401 (2009).